# Late-rising CD4 T cells resolve mouse cytomegalovirus persistent replication in the salivary gland

Simon Brunel[1]☯, Gaelle Picarda[1]☯, Ankan Gupta[1,2], Raima Ghosh[1], Bryan McDonald[1]¤, Rachid El Morabiti[1], Wenjin Jiang[1], Jason A. Greenbaum[3], Barbara Adler[4], Gregory Seumois[5], Michael Croft[6], Pandurangan Vijayanand[5], Chris A. Benedict[1]*

1 Center for Infectious Disease and Vaccine Research, Center for Autoimmunity and Inflammation La Jolla Institute for Immunology (LJI), La Jolla, California, United States of America, 2 Division of Immune Regulation, La Jolla Institute for Immunology (LJI), La Jolla, California, United States of America, 3 LJI Bioinformatics Core, La Jolla Institute for Immunology (LJI), La Jolla, California, United States of America, 4 Max von Pettenkofer Institute & Gene Center, Virology, Faculty of Medicine, Ludwig- Maximilians- University Munich, Munich, Germany, 5 Center for Cancer Immunotherapy, Center for Autoimmunity and Inflammation, La Jolla Institute for Immunology (LJI), La Jolla, California, United States of America, 6 Center for Autoimmunity and Inflammation, La Jolla Institute for Immunology (LJI), La Jolla, California, United States of America

☯ These authors contributed equally to this work.
¤ Current address: Department of Molecular Biology, University of California San Diego, San Diego, California, United States of America
* benedict@lji.org

**Data Availability Statement:** The data discussed in this publication have been deposited in NCBI's Gene Expression Omnibus and are accessible through GEO Serie accession number GSE237946.

## Abstract

Conventional antiviral memory CD4 T cells typically arise during the first two weeks of acute infection. Unlike most viruses, cytomegalovirus (CMV) exhibits an extended persistent replication phase followed by lifelong latency accompanied with some gene expression. We show that during mouse CMV (MCMV) infection, CD4 T cells recognizing an epitope derived from the viral M09 protein only develop after conventional memory T cells have already peaked and contracted. Ablating these CD4 T cells by mutating the M09 genomic epitope in the MCMV Smith strain, or inducing them by introducing the epitope into the K181 strain, resulted in delayed or enhanced control of viral persistence, respectively. These cells were shown to be unique compared to their conventional memory counterparts; producing higher IFNγ and IL-2 and lower IL-10 levels. RNAseq analyses revealed them to express distinct subsets of effector genes as compared to classical CD4 T cells. Additionally, when M09 cells were induced by epitope vaccination they significantly enhanced protection when compared to conventional CD4 T cells alone. These data show that late-rising CD4 T cells are a unique memory subset with excellent protective capacities that display a development program strongly differing from the majority of memory T cells.

## Author summary

Functional memory T cells are critical to ultimately control chronic viral infection. We show that cytomegalovirus (CMV) induces a population of unique memory CD4 T cells

(https://www.ncbi.nlm.nih.gov/geo/query/acc.cgi?acc=GSE237946).

**Funding:** This work was supported by NIH grants R01AI101423 and R01AI139749 to C.A.B. The funders had no role in study design, data collection and analysis, decision to publish, or preparation of the manuscript.

**Competing interests:** The authors have declared that no competing interests exist.

that expand late during infection and resolve viral persistence much better than conventional T memory cells. Mutation of the genomic CMV epitope recognized by these T cells results in markedly enhanced persistent replication, and introduction of the epitope into a CMV strain that lacks it induces much quicker resolution of persistence. When induced via vaccination, we show these T cells provide significantly enhanced protection against acute viral challenge. Together our study identifies a novel population of CD4 T cells that utilize distinct effector function(s) to rapidly resolve persistent viral replication.

## Introduction

CD4 T cells coordinate host defenses by 'helping' both CD8 T cell [1,2] and antibody responses [3], as well as by mediating direct antiviral activity. They respond uniquely to the specific inflammatory cues and antigen presenting cells (APC) activated by diverse pathogens, allowing for significant flexibility in their subsequent differentiation and effector functions [4]. Consequently, the memory T cells that develop play a critical role in curbing chronic viral diseases such as HIV [5] and HCV [6,7] but also dampen replication and reactivation of persistent viruses that are typically less pathogenic when immunity is intact, such as the herpesviruses. In this regard, mounting a rapid and robust memory CD4 T cell response strongly correlates with protection against cytomegalovirus (CMV/HHV-5, a ß-herpesvirus) disease in both transplant and congenital infection [8–11]. This is also the case for rhesus CMV infection in monkeys [12]. Memory T cells are typically primed and expand/differentiate during the initial days of acute viral infection. For CD8 T cells, memory precursors are thought to develop during the first week of infection, arising in parallel with the effector T cell pool [13]. In the case of CD4 T cells, requirements for memory precursor development are less clear, and defined lineages such as regulatory (Treg), follicular helper (Th) and tissue resident (Trm) cells likely have unique requirements [14–17]. Much of this information has been gleaned from studies of Lymphocytic choriomeningitis virus (LCMV), which replicates at high systemic levels during chronic infection, promoting memory T cell disfunction like occurs in HIV and HCV infection [18]. In contrast, after a short, systemic acute replication phase, cytomegalovirus establishes selective persistence in the ductal epithelia of a few mucosal tissues to facilitate its secretion and horizontal transmission. Memory CD8 T cell exhaustion is not commonly observed during CMV infection, despite the fact that the virus is harbored for life and are periodically exposed to their cognate antigens causing them to undergo "memory inflation" [19]. Therefore, CMVs unique lifestyle induces distinct memory T cell differentiation as compared to other, more pathogenic chronic infections.

MCMV induces a robust and diverse CD4 T cell response early during infection in mice [20,21], with persistent replication in the salivary gland (SG) for several months, and CD4 T cells are absolutely critical for ultimately resolving this extended replication prior to "whole host latency" being established [22,23]. In humans, CMV can be shed from mucosal sites for years (e.g. SG, kidneys and breast), and our recent work has shown that the CD4 T cell response in people is equally broad and diverse [24]. Although, CD4 T cells are critical, the exact nature of the population that provides protection is not currently known. Host-produced IL-10 helps sustain mouse CMV persistence in the SG mucosa [25], primate CMVs encode their own IL-10 orthologues [26] and human CMV-specific CD4 T cells can produce IL-10 [27], indicating a key role for this immunosuppressive cytokine. Moreover, while IFNγ produced by conventional CD4 T memory cells that expand early during infection can dampen CMV persistence, this is not sufficient to fully curtail replication [25,28,29]. This again suggests

that the dynamics and nature of memory T cell development affording protection to CMV differs from that of other viruses. Previously we identified fifteen immunodominant epitopes recognized by CD4 T cells infected with the MCMV Smith strain [20]. Interestingly, while we found that T cells recognizing 14 of these epitopes displayed a conventional effector to memory transition profile with peak expansion at day 8 followed by a contraction phase by ~ 3 weeks, one population directed against an epitope derived from the viral M09 protein (M09[133-147]) only accumulated significantly at later times. Together these data suggested that M09-reactive CD4 T cells might be uniquely important for resolving CMV persistence. Here we show that these late-rising M09 cells develop and accumulate in the SG at high levels, long after conventional memory is established, and compose > 10% of all antiviral CD4 T cells. Most importantly, we show they are critical for resolving persistent viral replication, and also provide qualitatively better protection when induced through vaccination, highlighting them as a unique CMV memory subset.

## Results

### Expansion of late-rising M09 epitope-specific CD4 T cells parallel the resolution of MCMV persistent replication

CD4 T cells are critical for resolving persistent MCMV replication in the salivary gland (SG) [22,23], but the overall specificity, differentiation requirements and key effector functions of these T cells still remains largely unknown. In order to address this, we used a BAC-cloned isolate of the wild-type MCMV Smith strain (MCMV[Smith]) [30] in C57BL/6 mice (B6). Replication of MCMV[Smith] was detectable in the SG at day 8, increasing 10-fold by day 15 and remaining unchanged through day 40. Replication was reduced by 5-fold at day 50 of infection and was completely controlled in more than 70% of mice by day 55 (**Fig 1A**), suggesting the memory CD4 T cells required for resolving persistence take an extended time to differentiate. From the 15 epitope-specific CD4 T cell responses we identified previously in MCMV[Smith] infected B6 mice [20], 14 were detectable in the spleen during the first week of infection upon epitope restimulation. Notably one response directed against an epitope derived from the viral M09 protein (M09[133-147], *M09 hereafter*), only expanded to significant levels at later times [20]. To test whether the resolution of MCMV[Smith] persistence in the SG might coincide with the emergence of these "late-rising" M09-reactive cells, MHC-Il tetramers were generated to track their development. In addition, two tetramers specific for CD4 T cells reactive with epitopes derived from the viral M25 and M142 proteins (M25[409-423] and M142 [24–38], *M25 and M142 hereafter*), cells which show a conventional pattern of expansion and contraction during the first few weeks of infection, were also generated. These tetramers were used in flow cytometry to identify M09, M25 and M142 cells present in SG tissue at times between 8- and 200-days post infection (**Fig 1C left and right graph**). M09 cells were barely detectable on day 8 and 15 of infection, while the frequency and number of M25 and M142 cells peaked on day 8 and contracted thereafter. In contrast, a high frequency of M09-reactive T cells was detectable in the SG at day 40, persisting at this elevated number through day 55 when viral replication was resolved, whereas M25 and M142 cells were only present at very low numbers at these times (**Fig 1B and 1D**). Similar results were also seen in the spleen (**S1A and S1B Fig**) and in the blood (**S1C and S1D Fig**) for these three MCMV CD4 T cell populations. These data indirectly implied that M09 cells may promote the ultimate resolution of viral replication in the SG at times of later-persistence.

### The M09[133-147] epitope plays a critical role in MCMV persistence

To test this hypothesis directly, we analyzed the persistent replication of MCMV strain K181 in the SG. MCMV[K181] was originally isolated after serial passage of MCMV[Smith] in mice [31]

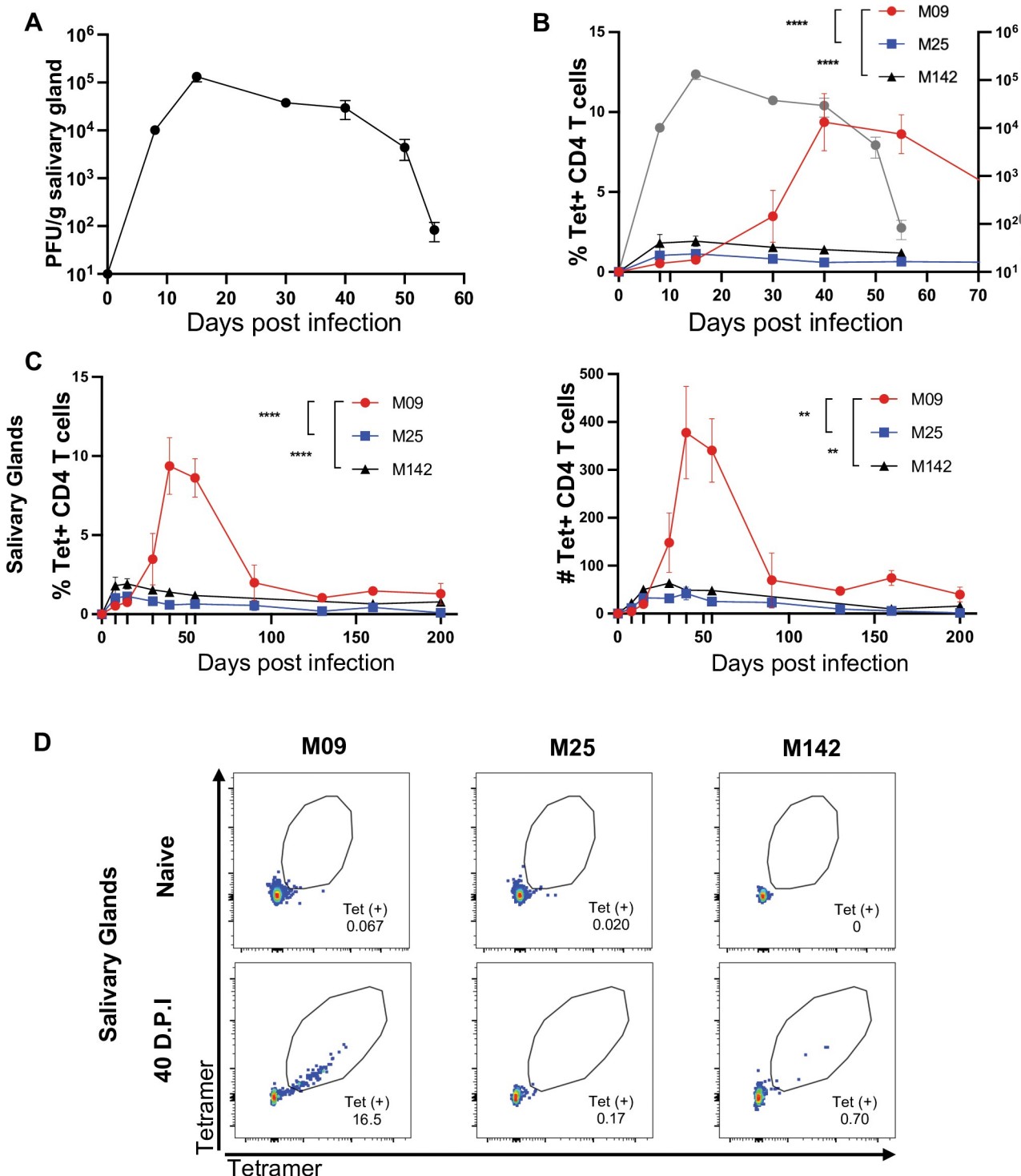

**Fig 1. Expansion of Late-rising M09 CD4 T cells coincides with the resolution of MCMV persistence in the salivary gland.** (A) Replication of MCMV [Smith] in the SG determined at various days post infection (dpi) by plaque assay. (B) Overlap of MCMV replication levels (A) (right Y axis) and frequency of MCMV epitope-specific CD4 T cells over time. (C) M09 (red), M25 (blue) and M142 (black) MHC-II tetramer-binding CD4 T cells in the SG from day 0 to day 200 post infection, in percentage (left) and absolute number (right). (D) Representative flow cytometry plot of M09, M25 and M142 CD4 T cells stained with tetramers labeled with 2 different fluorophores (PE and APC) in SG at 0 and 40 D.P.I. All time points represent 4–8 mice and experiments were repeated a minimum of two times. Statistical significance determined by Two-way Anova test is shown between M09 and M25 or M142. **p < 0.01, ****p < 0.0001.

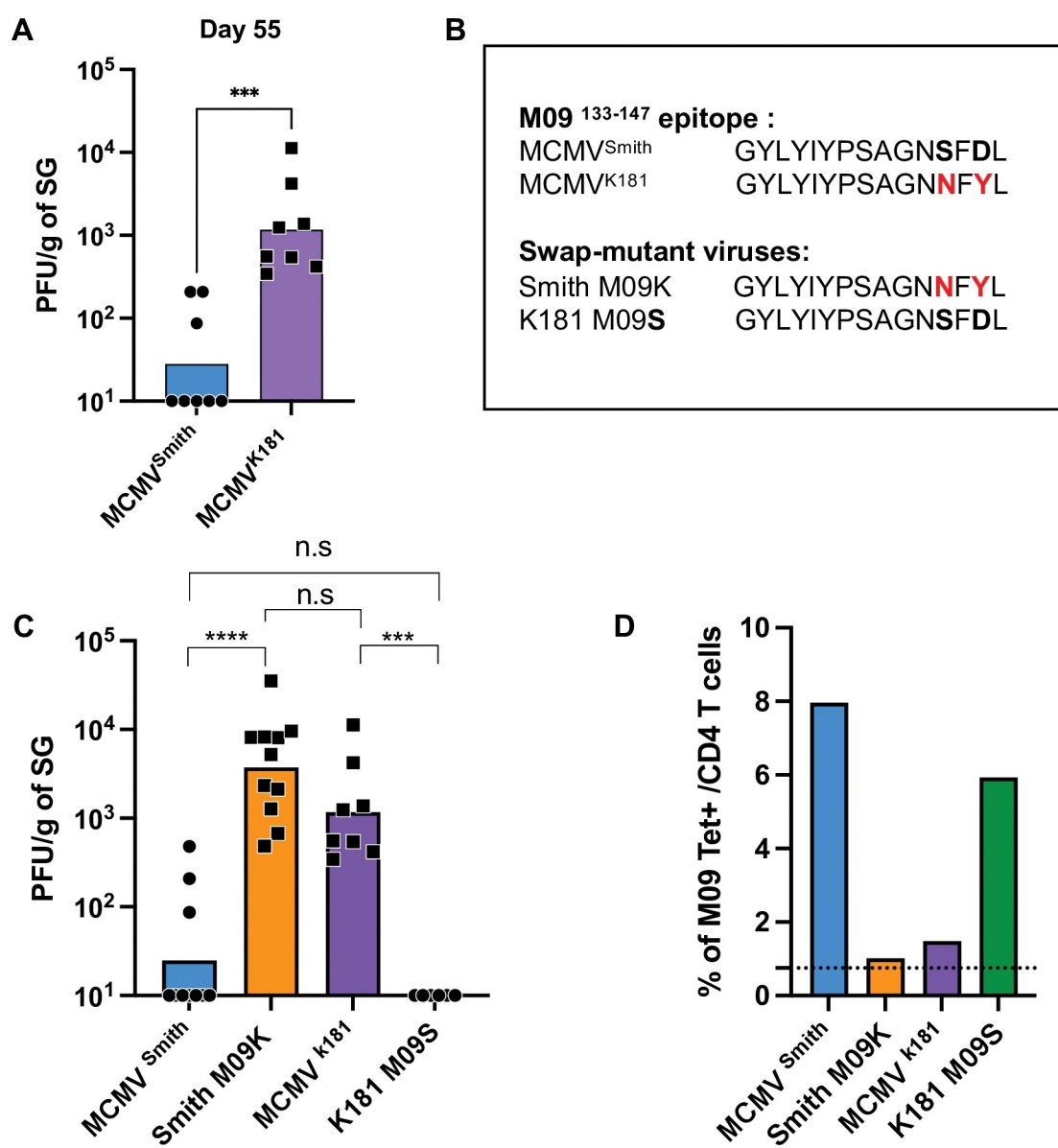

**Fig 2. The M09 [133–147] epitope plays a critical role in resolving MCMV persistence in the SG.** (A) MCMV[Smith] and MCMV[K181] replication levels at 55 dpi in the SG. (B) Amino acid sequence of the M09[133-147] epitope in the Smith and K181 wild-type strains, and mutations generated in the Smith M09K and K181 M09S 'swap mutants'. (C) Wild-type (Fig 2A) and epitope-swap mutant replication levels at 55 dpi in the SG. (D) Percentage of tetramer reactive M09 cells in the SG at 48 days of infection. Dashed line represents background of tetramer binding in uninfected mice. Statistical significance was assessed by Mann Whitney test, **p < 0.01, ****p < 0.0001.

in order to try and select for a viral variant displaying enhanced pathogenesis. Consistent with this, we found MCMV[K181] replicated for an extended time compared to MCMV[Smith] in the SG (**Fig 2A**). Notably, when sequences of all 20 MCMV epitopes identified to be targeted by CD4 T cells in B6 mice were compared in these two strains [32], the M09[133-147] epitope was only one of three (other 2 are M14[136-150] [33] and M45[192-206], both that show normal expansion kinetics) found to vary in sequence (**Fig 2B**), suggesting the enhanced persistence of MCMV[K181] might be due to an inability to induce M09 T cells [20,32,33]. To test this hypothesis, we generated "epitope-swap" mutant viruses where the M09 epitope in MCMV[Smith] was

interchanged with the analogous sequence from MCMV[K181] (Smith M09K), and another mutant virus where the M09 epitope from MCMV[K181] was interchanged with the analogous sequence of MCMV[Smith] (K181 M09S) (**Fig 2B**). Smith M09K and K181 M09S replicated identically to the wild-type strains in the spleen and liver at day 4 (**S2A and S2B Fig**), indicating that mutations of the M09 epitope did not globally impair their fitness. The level of replication in the SG at day 55 for MCMV[K181] and Smith M09K was comparable, and ~5x log[10] higher than MCMV[Smith] (**Fig 2C**). Additionally, all mice infected with K181 M09S resolved viral persistence by day 55, whereas no mice infected with MCMV[K181] showed control at this time. In turn, no mice infected with Smith M09K showed resolution of persistence by day 70 and MCMV[K181] still showed high levels of persistence at day 130 post infection (**S2C and S2D Fig**). As expected, both MCMV[K181] and Smith M09K K181 did not induce significant levels of M09 cells, while MCMV[Smith] and K181 M09S infected mice showed high frequencies (**Fig 2D**). Taken together, these data clearly show that late-rising M09 CD4 T cells, which differ from other conventional MCMV CD4 T cells that expand early during infection, play a key and non-redundant role in resolving persistent viral replication in the SG.

## Late-rising CD4 T cells are better cytokine producers than their conventional counterparts during MCMV persistent infection

To investigate the phenotype of these protective M09 T cells, we assessed their effector cytokine production. It is known that CD4 T cell-derived IFNγ plays an important role in the resolution of MCMV persistence [28,29]. Therefore, we compared the relative production of IFNγ and IL-2 by M25 and M09-reactive cells in the SG at day 8 and day 40, the times when each population peaked in numbers (**Fig 3A**). At day 8, 30% of SG-resident M25 cells produced IFNγ following restimulation, significantly more than at day 40 (4%). In contrast, 40 days post infection ~50% of M09 cells produced IFNγ. Additionally, no M25 cells produced IL-2 at either day 8 or day 40 in the SG, while ~25% of M09 cells did (**Fig 3A**). We then examined M09, M25 and M142 cytokine production in the spleen at day 8 to 200 of infection (**Fig 3B–3F**). M09 cytokine production (IFNγ, TNF and IL-2) was barely detectable on day 8 and day 15, while that of M25 and M142 cells peaked on day 8 and dropped significantly by day 15. In contrast, a significant number of M09-reactive, cytokine producing T cells were detectable from day 40 to 200 days post infection. Additionally, the proportion of M09 cells that produced 2 or 3 of these cytokines at day 40 was significantly higher than M25 or M142 at day 8 or 40 (**Figs 3F and S3E–S3H**), and the level of cytokine production per M09 cell was also higher (e.g MFI, **S4 Fig**). Those data suggest that the late-rising M09-reactive memory subset is a better antiviral population compared to the conventional M25 or M142-reactive memory T cells at during MCMV persistence.

## Phenotypic and transcriptomic analyses of M09 T cells

To further explore potential phenotypic differences between late-rising M09 and conventional M25 T cells, expression of several cell surface markers was assessed in tetramer binding, SG-resident cells (**Figs 4 and S3**). At day 8, M25 cells highly expressed CD69 and the differentially glycosylated isoform of the activation marker CD43 recognized by the 1B11 antibody (CD43 hereafter) [34]. Expression of CD27, which is downregulated in some HCMV-specific CD4 T cells during latency [9,35,36], was largely similar to that of naive CD4 T cells at day 8 of infection (**Fig 4A**). However, at day 40, the peak of M09 cell expansion, the majority of these late-rising cells showed low CD43 and CD27 expression, but had similar CD44 and CD69 levels to that of M25 cells (**Fig 4A**). Notably, many other cell-surface activation/differentiation markers were expressed similarly in M09 and M25 cells at day 40 (**S5 Fig**). In addition, polyclonal

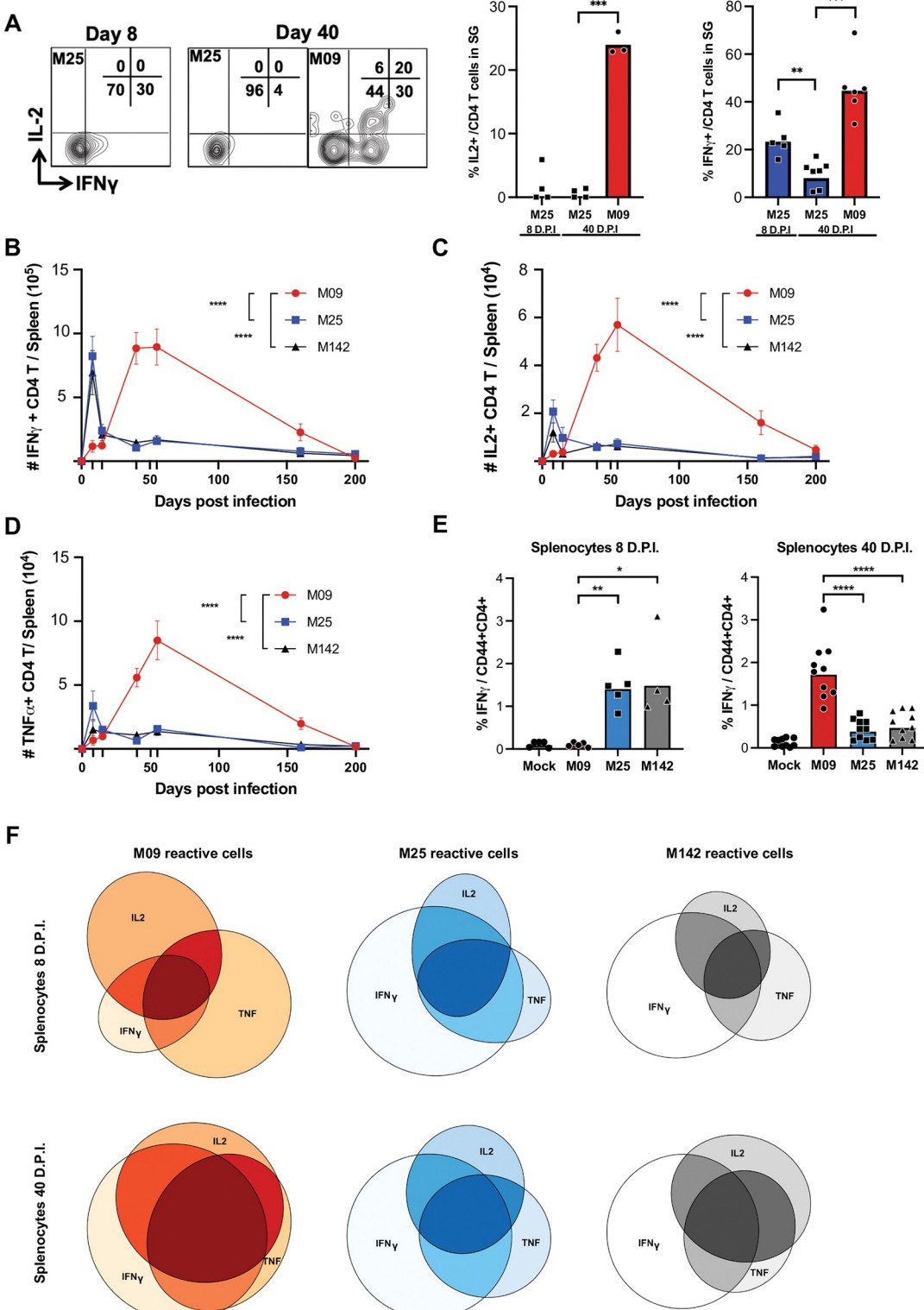

**Fig 3. Late-rising CD4 T cells are better cytokine producers during viral persistence.** (A) Representative flow cytometry plot for IFNγ and IL-2 expression by SG resident M09 and M25 reactive cells following *ex vivo* peptide restimulation. (B) Absolute number of IFNγ (B), IL-2 (C) and TNFα (D) producing splenic CD4 T cells from day 8 to 200 of infection. (E) Percentage of CD4 memory (CD44+) splenocyte expressing IFNγ after peptide stimulation by M09, M25 or M142 peptide at day 8 (left) and day 40 (right) post infection. (F) Euler representation of the poly-functionality of M09, M25 and M142 reactive

CD4 T cells from the spleen at day 8 (upper row) and day 40 (lower row) after MCMV infection. Kinetic differences were determined by Two-way Anova test. ****p < 0.0001. Group comparisons were assessed by Mann Whitney test, *p< 0.1**p< 0.01, ***p<0.001, ****p<0.0001.

MCMV CD4 T cells showed an expression profile for CD43 and CD27 similar to M25 cells at day 8, while at day 40 they showed a phenotype intermediate to M09 and M25 cells (**Fig 4A**). To further compare M09 and M25-reactive T cells, both were sorted at day 40 based on their differential expression of CD43 and analyzed by RNAseq. Notably, IL-10 and several positive upstream inducers of this cytokine were expressed at higher levels in M25 CD43$^{Hi}$ cells as compared to M09 CD43$^{Lo}$ cells (**Fig 4B and 4C**).

To assess IL-10 protein expression in SG-resident CD4 T cells, IL-10$^{GFP}$ reporter mice were infected with MCMV and the absolute numbers of IL-10$^{GFP+}$ M25 and M09 cells were determined at day 40. IL-10$^{GFP+}$ M25 and M09 cells were present at similar numbers at this time of viral persistence (40 and 50 cells/SG, respectively) (**Fig 4D**), despite the fact that ~100 fold more IFNγ -producing M09 cells were seen at this time (**see Fig 3B**). Additionally, the minor population of CD43$^{Hi}$ M09 cells were found to express more IL-10 than those that were CD43$^{Lo}$, suggesting a distinct differentiation program for these two subsets (**Fig 4E**). Based on these results, we postulated that low expression of CD43 could be used as a surrogate marker for the M09 population of MCMV CD4 T cells, and potentially other M09-like antiviral CD4 T cells, that provide better control of viral persistence due to high IFNγ expression and low IL-10 production. To test this, total virus specific CD4 T cells [37,38] were isolated at day 40, sorted based on CD43 expression, transferred into naive mice and subsequently challenged with MCMV. Strikingly, only CD43$^{Lo}$ cells provided some control of MCMV replication in the SG at day 15, with CD43$^{Hi}$ cells mediating no control (**Fig 4F**). These results indicate that low expression of CD43 marks the majority of late-rising M09-reactive T cells that are better at resolving viral persistence.

## Vaccine-induced M09 cells are highly protective

To further test whether late-rising M09 cells could play a unique and non-redundant role in resolving MCMV persistence, we asked whether they might provide qualitatively better control than conventional MCMV CD4 T memory cells when induced by vaccination. Mice were vaccinated with three 15mer epitopes that display a conventional effector-memory transition during acute infection (M25+M139+M142,'3conv') [20], with or without the M09 epitope. Importantly, a single peptide (M09, M25 or M142) vaccination regiment expanded equivalent numbers of cells in the spleen prior to virus challenge (**Fig 5A**), consistent with our observations that similar numbers of CD4 T precursors for these 4 populations exist in naive B6 mice. However, surprisingly upon MCMV challenge, mice vaccinated with any of these regiments showed no reduction in SG replication levels measured at day 15 (**Fig 5B**). Suggesting this was not because the immunization had been ineffective, we found that vaccinating with two conventional CD4 T epitopes (M25 and M142) provided protection against MCMV challenge in the liver, equivalent to immunizing with two characterized CD8 T cell epitopes (M38$^{316-323}$ + IE3$^{416-423}$) [39]. Strikingly, when the M09 peptide epitope was added to the M25+M142 vaccination regiment, 75% of mice completely controlled viral replication in the liver by day 8 (**Fig 5C**). Given the importance of IL-10 in sustaining MCMV SG replication, and the fact that it is highly expressed by CD4 T cells at times of early-persistence, we suspected this cytokine might restrict the ability of vaccine-induced CD4 T cells to exert control in this tissue. Consequently, vaccination was repeated in combination with antibody blockade of IL-10R signaling (αIL-10R), with protection measured at day 8 and 15 following MCMV challenge. In the

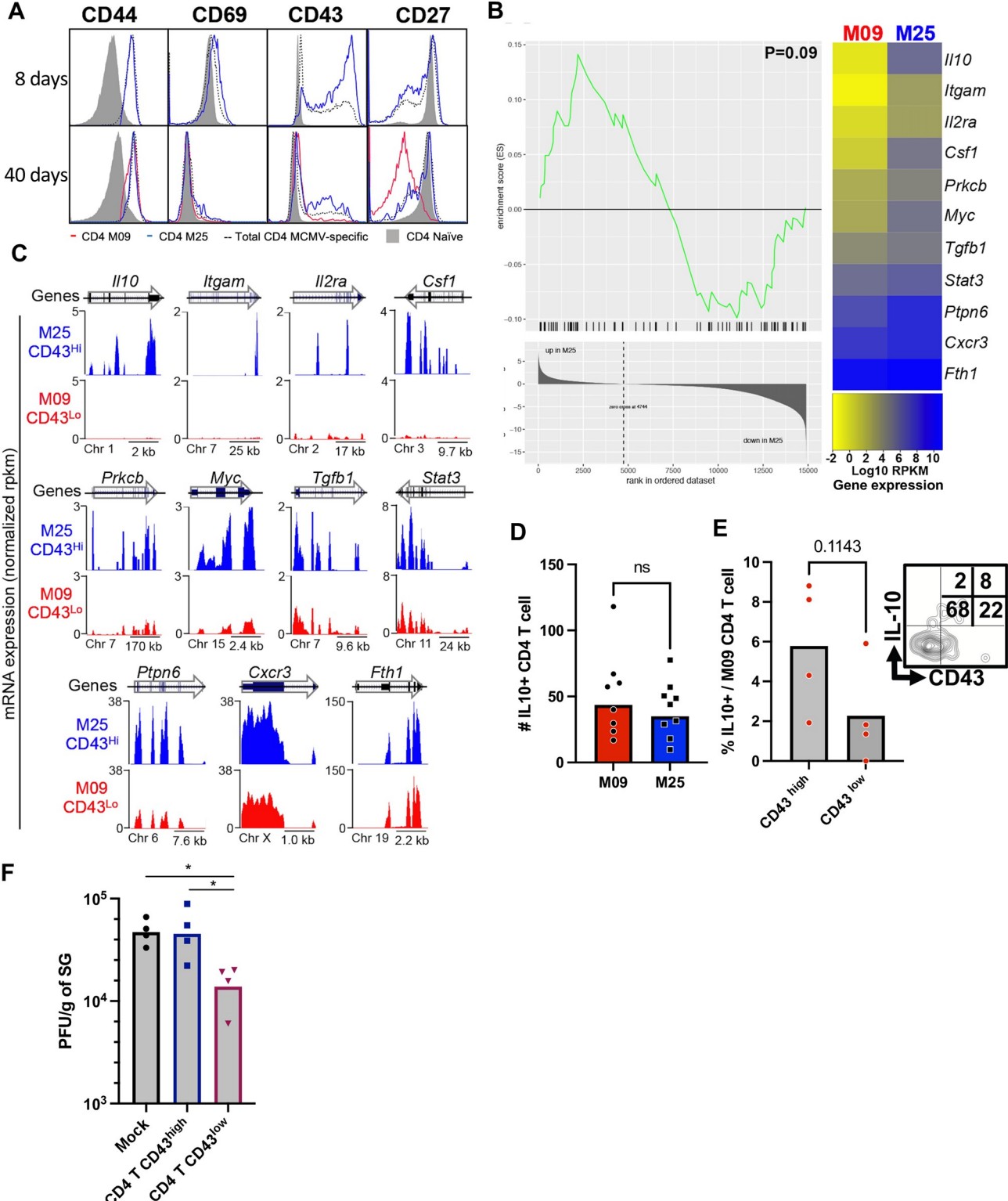

**Fig 4. Cell surface and transcriptomic phenotypes of M09 and M25 CD4 T cells.** (A) Cell surface phenotypes of conventional (M25, blue) and late-rising (M09, red) tetramer binding CD4 T cells from the SG at day 8, 15 and 40. Total MCMV-specific (CD11a^hiCD49d+) (dashed) and naïve (CD11aloCD49d-) (gray filled histogram) CD4 T cells are also shown. (B) M09-CD43^lo and M25-CD43^hi cells sorted from spleens of day 40 infected mice were subjected to RNA-seq. As IL-10 mRNA levels were found to be low in M09 cells, IPA analyses of core IL-10 inducers was performed followed by gene set enrichment analyses (GSEA). The heatmap shows the IL-10 regulating genes with differences of P<0.2. (C) Bulk RNA-Seq enrichment of

expression along genomic coordinates for genes listed in Fig 4B using UCSC genome tracks. Arrows indicate gene orientation and lines within arrows are exons. RPKM, reads per kilobase per million mapped. (D) Absolute number of IL-10[GFP+] M09 and M25 SG-resident cells at 40 days of infection. (E) IL-10 expression by CD43hi/lo M09 tet+ CD4 T cells at 40 D.P.I. (F) Splenic total MCMV-specific (CD11a[hi]CD49d+) CD4 T cells were sorted for CD43[Hi] and CD43[Lo] expression at day 45 of infection, 5e4 cells were adoptively transferred into naïve mice, subsequently challenged with MCMV and SG replication was determined at day 15. Statistical significance was assessed by Mann Whitney test, *p<0.1.

presence of αIL-10R, the results now largely recapitulated what was seen in the liver, with the 3conv+αIL-10R regiment providing ≥ 20-fold protection compared to 3conv or αIL-10R treatment alone (**Fig 5D**). Most notably, inclusion of the M09 peptide with IL-10 blockade completely controlled MCMV SG replication in ≥ 50% of mice SGs at both time points (**Fig 5D**). Taken together, these data show that inducing early expansion of normally late-rising M09 memory CD4 T cells through vaccination provides dramatically better protection against viral persistence compared to conventional memory CD4 T cells alone.

## Discussion

This work shows that an epitope-specific CD4 T cell response that expands late during MCMV infection is dramatically better at curtailing persistent replication during both natural infection and vaccination when compared to conventional memory cells that develop much earlier. MCMV induces several epitope-specific memory CD8 T cell responses that expand slowly and/or don't contract, termed 'memory inflation' [40–42]. However, M09-specific cells are the only known herpesvirus-specific CD4 T cells not efficiently primed during the first weeks of infection, although examples of this do exist in polyoma virus and LCMV infection [43,44], and therefore we have defined them as 'late-rising'. Development of MHC-II tetramers allowed us to characterize these late-rising T cells in the SG tissue, the main site of CMV persistence and dissemination, revealing them to produce much less IL-10 and more IFNγ and IL-2 than conventional virus-specific CD4T cells at this site. It also allowed us to perform the presented RNAseq experiments. Additionally, these late-rising cells provide markedly better protection as compared to conventional memory CD4 T cells when co-induced via peptide vaccination. In total, the results support a model where CMV infection drives the differentiation of a unique, highly protective subset of late-rising T cells that can better resolve persistence as compared to conventional or tissue-resident antiviral memory cells that arise early during MCMV and other persistent infections [15,45,46].

The K181 strain has two amino acids which differ in the Smith M09 epitope, and this facilitated generating M09 epitope-swap mutants, formally demonstrating the non-redundant role of these late-rising cells in resolving MCMV persistence [20,21,36]. It is not precisely clear how K181 was derived from serial passage of Smith in the 1960s [31,32,47], but the possibility exists that K181 is an M09 'escape mutant'. However, significant genomic differences between Smith and K181 exist, suggesting the original Smith isolate also may have contained both strains [48] and several wild MCMV strains also vary in the M09 15mer epitope [32]. Consequently, the precise origin of K181 is unclear, but a reasonable hypothesis is that selective pressure against inducing late-rising M09 cells may exist. There is some precedent for the emergence of CD4 T cell epitope escape mutants that enhance chronic LCMV infection, reported in immune compromised mice or with transgenic SMARTA CD4 cells [49]. However, to our knowledge, this study is the first example where a population of memory CD4 T cells having such a dramatic impact on viral persistence shows apparent epitope-selection during natural infection.

M09 cells did not expand to higher levels on day 20 when administering αIL-10R from the time of initial infection, suggesting IL-10 does not regulate their late-rising phenotype, albeit

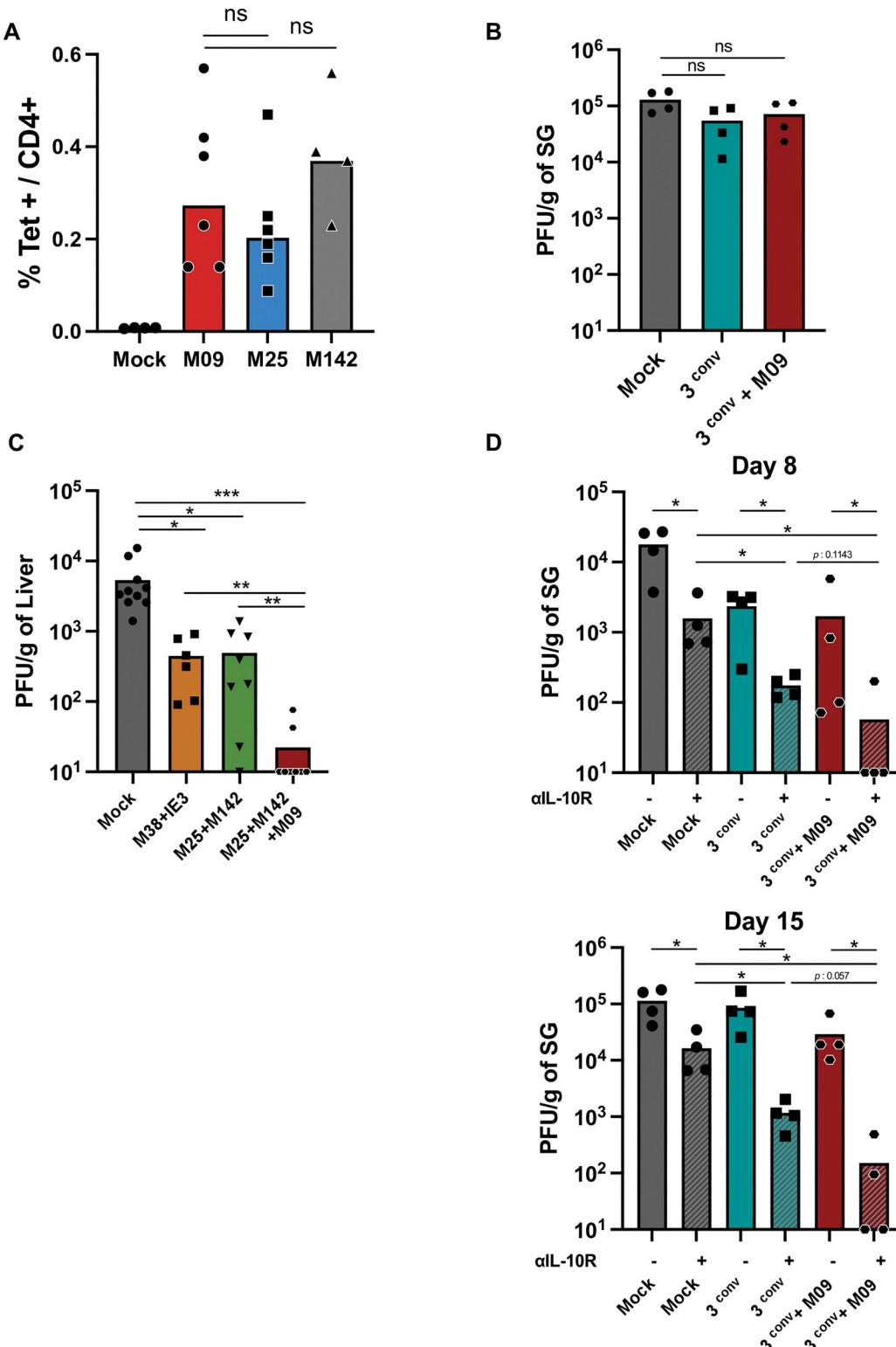

**Fig 5. Vaccine induced M09 cells provide dramatic protection.** (A) Mice vaccinated by a single immunization (CFA) with M09, M25 or M142 peptides alone were analyzed for their numbers of tetramer positive cells in the spleen prior to MCMV challenge. (B) Mice vaccinated with M25+M142+M139 peptides (3conv) or 3conv+M09 were challenged with MCMV[Smith], and SG replication levels were determined at day 15. (C) Mice were vaccinated with M25+M142, M25 +M142+M09 or two CD8 T cell epitopes (M38+IE3), and liver replication levels were determined 8 days post MCMV

challenge. (D) Blocking anti-IL-10R antibody was injected at day 0 and 4 (for d8) or at day 0 and 7 (for d15) in vaccinated mice, and SG replication levels were determined at day 8 (upper panel) or 15 (lower panel) after MCMV challenge. Statistical significance was assessed by Mann Whitney test, ***p< 0.001, **p< 0.01, *p<0.1.

IL-10 deficient mice show enhanced expansion of other MCMV-specific CD4 T cells [50]. Another possibility is that the M09 epitope is not efficiently presented early during infection, or is presented by an APC which doesn't foster their robust expansion. These cells can be recalled early during infection, as shown by our vaccination studies, but the requirements for expansion of naïve and memory cells is clearly different. Also, the precursor frequencies of M09, M25 and M142 are likely similar based on Fig 5A, reflecting measurements in naïve mice. MHC-II levels are reported to be low on MCMV-infected SG epithelial cells [28], raising the question of how antiviral CD4 effector T cells ultimately control vial persistence in this organ. However, this was analyzed at 3 weeks of infection, a time where IL-10 is highly produced by conventional CD4 T memory cells and M09 cells are far from peak levels, and IL-10 suppresses MHC-Il during MCMV infection [51]. High IFN-I levels are produced by stromal and DC populations during acute MCMV infection [52], which induces IL-27 that subsequently promotes IL-10 production by MCMV CD4 T cells [53], so perhaps lower IFN-I levels during late-persistence renders this axis less dominant. We hypothesize that M09 cells mediate direct, and potentially unique, effector functions in the SG to orchestrate the resolution of persistence. This may include CTL activity, as we have shown in the liver of MCMV infected mice [54]. Moss and colleagues showed that HCMV epitope-specific CD4 T cells can exhibit CTL activity [36], and recent results suggest gB-specific CD4 T can eliminate HCMV infected senescent fibroblasts [55]. Notably, CD43+ CD8 T memory cells show enhanced secondary expansion in other infection models [56,57], and we are currently assessing this for MCMV-specific CD4T cells. As well as highlighting the nature of these protective late-rising T cells induced by natural infection, our data represent the first report of a CD4 T cell-centric CMV vaccine strategy. This was attempted because mounting a robust CD4 T cell response reduces the duration of HCMV shedding and may restrict congenital infection [58,59], and depleting CD4 T cells promotes congenital CMV infection in rhesus monkeys [60]. We believe future CMV vaccine strategies should carefully assess the phenotype of induced CD4T cells, an underexplored area in past clinical trials [61,62]. The presence of late-rising CD4 T cells may be specific in the case of MCMV, but we would posit that similar cells arise in monkeys and humans in order to eventually control CMV persistence. Merely looking in the peripheral blood of HCMV+ individuals at virus epitope specific CD4 T cells would not answer this question, as in most cases infection occurred several years ago, but including a kinetic analysis in vaccinated or transplanted HCMV- cohorts could start to assess this. CD4 T cell vaccination can induce immune pathology in chronic CMV infection [63], but this is likely due to very high levels of systemic replication and immune cell activation in this model as compared to MCMV [64]. Interestingly, CD4 T cell depletion does not alter MCMV liver replication levels at day 8, indicating they are unnecessary or redundant at this time, further highlighting their striking vaccination protective capacity. Why co-blockade of IL-10 is required in the SG but not the liver for CD4 T cell vaccine protection is intriguing, and is supported by reports of a redundant role for IL-10 in MCMV control in the liver [65]. It is likely that the distinct cell-types infected and the unique immune environments in these two organs impact the importance of IL10 suppression.

In summary, late-rising M09 CD4 T cells both resolve viral persistence during natural infection and protect better in vaccination, likely due to their unique phenotype and effector functions as compared to conventional memory CD4 T cell responses.

## Materials and methods

### Ethics statement

This study was carried out in strict accordance with the guidelines of Association for assessment and Accreditation of laboratory Animal Care (AAALAC) and National Institutes of Health (NIH). All animal protocols used in this study, were approved by the Institutional Animal Care and Use Committee (IACUC) of the La Jolla Institute for immunology (LJI).

### Mice, viruses and cells

Seven to 12wk old female wild-type C57BL/6 and IL-10 GFP reporter mice [66] were used for all experiments. Mice were bred under specific pathogen-free conditions in the Department of Laboratory Animal Care at LJI. Unless otherwise mentioned; mice were infected i.p. with $1x10^6$ PFU of MCMV BAC-derived strains generated in fibroblasts. MCMV virus stocks were generated by electroporating BAC DNA into 3T3 cells, subsequent expansion in MEFs, and were quantified by standard plaque assay in 3T3 cells [67].

Mutation of the MCMV[Smith] BAC was performed in E. coli by en-passant (ET) mutagenesis as described [68], and successful mutagenesis was confirmed by PCR sequencing. The entire BAC was also sequenced to verify no additional mutations existed in the Smith M09K mutant as compared to the Smith[WT] BAC. Fifty bp paired-end short reads from BAC clones were mapped against BAC vector sequence and E.coli (DH10B strain) as filtering via Bowtie 2.0.0 [69]. Unaligned reads were then mapped against MCMV sequences. Freebayes (v0.9.18-3-gb72a21b) was employed to call SNPs on the basis of the sorted BAM files. Vt package [70] was performed on filtered variants for normalization, followed by decomposing multiallelic variants as well as biallelic block substitutions into their constituent SNPs. The sequence was deposited in GenBank (KY348373). MCMV organ replication levels were determined by plaque assay in 3T3 cells as described [71].

### MHC-II tetramer staining and enrichment

Biotin-labeled M25 or M09 (1-A˚) monomers were generated by the NIH Tetramer Core Facility (Emory, AL) and were tetramerized in our lab by addition of streptavidin-bound fluorochrome according to their instructions. Single-cell suspensions were subjected to specific CD4 T cell tetramer enrichment essentially as described [72], or were directly stained with tetramers bound to both APC and RPE. Briefly, isolated mononuclear cells were incubated with fluorochrome-conjugated class II tetramer (6 ug/ml) for 120 min at room temperature (RT), then cells were stained with other flow cytometry antibodies or for enrichment anti-fluorochrome magnetic beads (Miltenyi-Biotec) were added as recommended by provider protocol. Bead-bound cells were then passed through magnetic separation (MS) columns, resulting in ~100-fold enrichment of tetramer-bound CD4 T cells.

### Cell isolation and flow cytometry

Single-cell suspensions of SGs were prepared from uninfected or MCMV-infected mice. SG-associated lymph nodes were carefully removed before digestion of tissue with RPMI media (+10% FBS) containing collagenase D (1mg/ml), DNase (0.1mg/ml) and CaCl2 (5mM). Digested samples were passed through a 70uM cell strainer and tissue associated lymphocytes were isolated by density gradient centrifugation using lympholyte-solution (Cedarlane). Splenocytes were isolated by gently mashing spleens through a 70uM cell strainer followed by addition of red blood cell lysis buffer (Thermofisher/eBioscience). Blood was collected by submandibular skin puncture, the volume of blood collected was precisely measured and red

blood cells were eliminated by red blood cell lysis buffer, number of cells was assessed by using Precision Count Beads (Biolegend). Tetramer-enriched or total CD4 T cells were stained with mAbs specific for CD3 (clone 17A2), CD4 (clone RM4-5), NK1.1 (clone PK136), CD44 (clone IM7), CD11a (clone M17/4), CD49d (clone R1-2), CD43 (clone 1B11), CD27 (clone LG.3A10), CD69 (clone H1.2F3), CD40L(MR1), CD137/4-1BB (1AH2), CD160(CNX46-3), GITR (DTA-1), CD134/OX40 (X-86), CD272/BTLA (6F7), Ly6c (HK1-4), PSGL1 (2PH1), CD279/PD1 (29F.1A12). In some experiments, cells were also stained for TNFα (MP6-XT22), IFNγ (clone XMG 1.2) and IL-2 (JES6-5H4). Unless otherwise mentioned, antibodies were purchased from either BD Pharmingen (San Diego, CA), Thermofisher/eBioscience (San Diego, CA) or Biolegend (San Diego, CA). Data were collected on a LSRII flow cytometer (BD Biosciences) and analyzed using FlowJo software.

## Adoptive transfers

WT MCMV-infected mice were euthanized at day 45 of infection and splenic CD4 T cells displaying a CD11a+CD49d+ phenotype, were FACS sorted for CD43 hi vs lo populations. Post-sort, 50,000 CD4 T cells were intravenously injected into naive mice 24 hours before MCMV challenge.

## Intracellular cytokine staining

Total mononuclear or tetramer-enriched cells were restimulated ex vivo with a combination of phorbol myristate acetate (PMA; 100 ng/ml; Sigma-Aldrich, USA), ionomycin (500 ng/ml; Sigma-Aldrich) and brefeldin A (2 ug/ml) for 5–6 hours at 37˚C. Following restimulation, cells were either antibody-stained directly or after tetramer- enrichment. Cells were then fixed and permeabilized (BD Pharmingen) before incubating with anti- IFNγ, TNFα or IL-2 antibody. For IL-10, Il10-GFP reporter mice (Vertex) were used.

## Vaccination

Mice were vaccinated with 50ug of each peptide in 50ul complete Freund's adjuvant (CFA) for priming and 50ul of incomplete Freund's adjuvant (IFA) for boosting 3 weeks later [73]. Mock-immunized mice received DMSO emulsified in CFA or IFA. Vaccinated mice were challenged 3 weeks following boost. For M09, M25 and M142 cells induction (Fig 5A), mice were vaccinated with 50ug of each peptide in 50ul complete Freund's adjuvant (CFA), spleen were analyzed 8 days later for tetramer positive cells.

## RNA-seq and GSEA

Viable tetramer positive M25 and M09 CD4 T cells were sorted from pooled spleens of infected mice based on expression levels of CD43 and CD27. Three pools of 5 mice were used. Total RNA was purified using a miRNA easy micro kit (Qiagen) and quantified then 50–1000 pg was amplified by the Smart-seq2 protocol and cDNA was purified using AMPure XP beads (1:1 ratio; Beckman Coulter). From this step, 1 ng cDNA was used to prepare a standard Nextera XT sequencing library (Nextera XT DNA sample preparation kit and index kit; Illumina). Samples were sequenced using a HiSeq2500 (Illumina) to obtain 50-bp single-end reads (Tru-Seq Rapid Kit; Illumina) generating in average of ~9 million mapped reads per sample. The reads were then aligned to UCSC mm10 reference genome using TopHat (v 1.4.1). After removing absent features (zero counts in all samples), the Raw counts were then Imported to R/Bioconductor package DESeq2 [74] to identify differentially expressed genes among samples. Values for differential expression are calculated using Wald test that estimates the

significance of coefficients in a fitted negative binomial generalized linear model (GLM). These p-values are then adjusted for multiple test correction using B. Hochberg algorithm [75] to control the false discovery rate. Differentially expressed genes between two groups of samples were defined as when DESeq2 analysis resulted in an adjusted P-value of <0.1 and the fold-change in gene expression was at least 2-fold. Principal component analysis (PCA) was performed using standard algorithms and metrics. Both whole-transcriptome amplification and sequencing library preparations were performed in a 96-well format to reduce assay-to-assay variability. Quality control steps were included to determine total RNA quality and quantity, the optimal number of PC preamplification cycles, and fragment size selection. Upstream positive regulators of IL-10 gene expression were identified using Ingenuity Pathway Analysis (Qiagen Bioinformatics). Pre-ranked gene set enrichment analysis (GSEA) was performed using log(P) * sign of fold-change as the ranking metric and 'classic' for the enrichment statistic [76].

The data discussed in this publication have been deposited in NCBI's Gene Expression Omnibus and are accessible through GEO Serie accession number GSE237946.

(https://www.ncbi.nlm.nih.gov/geo/query/acc.cgi?acc=GSE237946)

## Statistical analysis

Statistical significance was analyzed by suitable parametric (unpaired Student's t test with Welch's correction) or non-parametric test (Mann-Whitney U test). The difference in kinetic was determined by Two-way Anova test. Unless otherwise indicated, data was represented as individual value/mean ‡ SEM with p<0.05 (*), 0.01 (**), 0.001 (***), 0.0001(****).

## Supporting information

**S1 Fig. Analysis of MCMV epitope-specific CD4 T cells from spleens and blood.** (A) Representative flow cytometry plot of M09, M25 and M142 double tetramer stained CD4 T cells in spleens at 8 and 40 D.P.I. (B) M09 (red dot), M25 (blue square) and M142 (black triangle) specific tetramer CD4 T cells in the spleen from day 0 to day 200 post infection, in percentage (left) and absolute number (right). (C) Representative flow cytometry plot of M09, M25 and M142 dual color-stained tetramer of CD4 T cells in the blood at 14 D.P.I. Blood from naïve mice were stained with the same tetramers, a representative staining of M09 tetramer is shown. (D) Specific tetramer CD4 T cells in the blood from day 0 to day 200 post infection, in percentage (left) and absolute number (right). The difference in kinetic was determined by Two-way Anova test. ****p < 0.0001.
(TIF)

**S2 Fig. SG replication levels of MCMV[Smith], Smith M09K and MCMV[K181].** Replication of wild-type and Smith M09K in the spleen (A) and (B) liver at 4 days, and the SG (C) from day 30 to 70. (D) SG replication levels of MCMV[K181] (60–130 dpi), and both MCMV[Smith] and Smith M09K at 55 dpi. Statistical significance was assessed by Mann Whitney test for A and B, and C was determined by Two-way Anova test. **p < 0.01, ****p < 0.0001.
(TIF)

**S3 Fig. MCMV peptide specific reactive cells cytokine production during the infection.** Kinetics of the percentage of CD4 splenocyte expressing I IFNγ (A), IL-2 (B) and TNFα (C) after peptide stimulation by M09 (red dot), M25 (blue square) and M142 (black triangle) peptide from day 8 to 200 of infection. Level of expression of TNFα (E), IL-2 (F) or both (G) cytokines among IFNγ memory CD4 T cells (CD44+) at 40 dpi from splenocyte MCMV peptide stimulation. (H) Percentage of triple positive memory (CD44+) CD4 T cells cytokine producer

at day 40 (M09) and at day 8 (M25 and M142) post infection. (I) MFI of cytokine positive cells. Group comparisons were assessed by Mann Whitney test, *p< 0.1, **p< 0.01.
(TIF)

**S4 Fig. MCMV peptide specific reactive cells cytokine production during the infection (MFI).** (A) MFI of cytokine positive cells for IFNγ, IL2 and TNFα at day 8, day 40 and day 55 post infection. Group comparisons were assessed by Mann Whitney test, *p< 0.1, **p< 0.01.
(TIF)

**S5 Fig. Cell surface marker expression by MCMV-specific CD4 T cells.** Cell surface markers expressed by conventional (M25, blue) and late-rising (M09, red) tetramer-binding CD4 T cells in spleen is shown at day 8 or 40 of infection. Total MCMV-specific (CD11a^hiCD49d+, dashed histogram) and naïve (CD11a^loCD49d-, gray filled histogram) CD4 T cells are also shown.
(TIF)

## Acknowledgments

Flow Cytometry and Genomic sequencing core facilities at the La Jolla Institute.

Alec Redwood for providing the K181 BAC and discussions regarding MCMV epitope/ sequence variation.

## Author Contributions

**Conceptualization:** Simon Brunel, Gaelle Picarda, Chris A. Benedict.

**Data curation:** Simon Brunel, Gaelle Picarda, Ankan Gupta, Raima Ghosh, Bryan McDonald, Rachid El Morabiti, Wenjin Jiang, Chris A. Benedict.

**Formal analysis:** Simon Brunel, Gaelle Picarda, Ankan Gupta, Chris A. Benedict.

**Funding acquisition:** Chris A. Benedict.

**Investigation:** Simon Brunel, Gaelle Picarda, Chris A. Benedict.

**Methodology:** Simon Brunel, Gaelle Picarda, Chris A. Benedict.

**Project administration:** Simon Brunel, Gaelle Picarda, Jason A. Greenbaum, Gregory Seumois, Pandurangan Vijayanand, Chris A. Benedict.

**Resources:** Barbara Adler.

**Software:** Simon Brunel, Jason A. Greenbaum, Gregory Seumois, Pandurangan Vijayanand.

**Writing – original draft:** Simon Brunel, Gaelle Picarda, Jason A. Greenbaum, Chris A. Benedict.

**Writing – review & editing:** Simon Brunel, Michael Croft, Chris A. Benedict.

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
