## [Decision Letter · Decision Letter 0]

18 Sep 2023

Dear Chris,

Thank you very much for submitting your manuscript "Late-rising CD4 T cells resolve cytomegalovirus persistent replication" for consideration at PLOS Pathogens. As with all papers reviewed by the journal, your manuscript was reviewed by members of the editorial board and by several independent reviewers. This is a very interesting and well-conducted study that was received with interest by all the reviewers. However, in light of the reviews (below this email), we would like to invite the resubmission of a significantly-revised version that takes into account the reviewers' comments.

The primary aspect of the study that will need focus in a resubmission is whether M09-specific CD4 T cells are indeed functionally superior/different to other MCMV-specific CD4 T cells. Providing more insight into whether this is the case or not will substantially improve the manuscript and additional functional experiments and/or inclusion of exisiting data, as suggested by all 3 reviewers, will enable this. Also, although further transcription studies at different time-points are unnecessary, given the different kinetics of M25 and M09-specific T cells, some additional discussion regarding the possible limitations of the data presented in Figure 4 looking at a single time-point would be useful.

We cannot make any decision about publication until we have seen the revised manuscript and your response to the reviewers' comments. Your revised manuscript is also likely to be sent to reviewers for further evaluation.

Sincerely,

Ian R Humphreys

Guest Editor

PLOS Pathogens

Patrick Hearing

Section Editor

PLOS Pathogens

Kasturi Haldar

Editor-in-Chief

PLOS Pathogens

orcid.org/0000-0001-5065-158X

Michael Malim

Editor-in-Chief

PLOS Pathogens

orcid.org/0000-0002-7699-2064

This is a very interesting and well-conducted study that was received with interest by all the reviewers.

The primary aspect of the study that will need focus in a resubmission is whether M09-specific CD4 T cells are indeed functionally superior/different to other MCMV-specific CD4 T cells. Providing more insight into whether this is the case or not will substantially improve the manuscript and additional functional experiments and/or inclusion of exisiting data, as suggested by all 3 reviewers, will enable this. To compare cytokine expression (e.g., IFNg, TNF) between different antigen-specific T cells, you may also wish to consider analyzing the intensity of cytokine expression from your existing data (e.g., Fig 3) to get a sense of whether M09-specific CD4 T cells express more antiviral cytokines on a per cell basis as compared to other antigen-specific CD4 T cells.

Also, although further transcription studies at different time-points are unnecessary, given the different kinetics of M25 and M09-specific T cells, some additional discussion regarding the possible limitations of the data presented in Figure 4 looking at a single time-point would be useful.

Reviewer's Responses to Questions

**Part I - Summary**

Reviewer #1: In this manuscript, the authors investigate a population of memory CD4 T cells that undergo late-stage expansion during cytomegalovirus (CMV) infection and playing a crucial role in viral clearance. These CD4 T cells are specific to the viral protein M09, expanding after the conventional CD4 memory T cell response has already reached its peak and subsequently contracted. Notably, these late-expanding CD4 T cells exhibit distinct functional characteristics, demonstrating better ability to resolve viral persistence compared to conventional memory T cells.

The authors have conducted elegant experiments to test their hypotheses, yielding interesting and compelling results. While this manuscript is generally sound, there are a few remarks that the authors should consider addressing to further enhance its quality.

Reviewer #2: Brunel and colleagues have investigated the functional impact of late-rising CD4 T cells specific for the M09 epitope from MCMV. The data demonstrates that M09-specific T cells modulate the late control of MCMV in B6 mice and the authors show that this response underlies the differences in SG replication exhibited by Smith and K181 strains of MCMV in this model. Moreover, the authors show that M09-specific T cells can be targeted to improve viral control. These data are exciting and significant. The authors further suggest that the M09-specific T cells are effective because they are more functional than the T cells primed earlier in infection. In this reviewer's opinion, this last conclusion requires some more justification, or perhaps the discussion of the data should be more nuanced. I do think that the function is important, and that some further analyses of the current data would provide some more insights. However, regardless of the answers about T cell function, this study overall provides significant new insights into the MCMV model and immune control of CMV in general.

Reviewer #3: The manuscript by Brunel et al extends previous work on the role of CD4+ T cells in the persistence and subsequent control of Mouse cytomegalovirus (MCMV) infection in the salivary gland (SG) following primary infection.

The authors show that MCMV replication peaks in the SG at day 15 and then persists up to approx. 40 days before starting to decline and become controlled. Using MHC Class II MCMV tetramer reagents the kinetics of various MCMV specific CD4 T cells specific to a number of different peptides was determined, identifying T cells specific to a peptide in M09 which developed late in infection and was co-incident with the decrease in MCMV replication in the SG. The paper then seeks to demonstrate that the late control of MCMV is dependant on these M09 specific T cells, demonstrating that another strain of MCMV persistence in the SG was not controlled and this stain had a mutation in the specific peptide mapped to these T cell, this phenotype could be swapped between strains by swapping the peptide sequence which also affected if M09 specific T cell were or were not generated. M09 specific CD4 T cells contained a higher frequency of IFNg/TNFa producing cells as compared to T cells that were generated early in primary infection. A detailed Phenotypic and transcriptomic analysis of M09 v the early rising M25 was performed and demonstrated phenotypic differences as well as a difference In the frequency of IL-10 immunosuppressive v IFNg with M09 specify T cells being heavily skewed towards more anti -viral like cells. Adoptive transfer of these cells followed by MCMV challenge demonstrated better SG MCMV control. Finally, a vaccination strategy using peptides that generated early conventional epitopes (eg M25) v late M09 followed by MCMV challenge, the results from this analysis were not completely straight forward but broadly supported the conclusion that the M09 specific T cells were critical in brining persistent MCMV replication in the SG under control. The results explain the kinetics of MCMV persistence and resolution and describe a interesting difference in the properties of immediate and late rising CD4 T cells.

**Part II – Major Issues: Key Experiments Required for Acceptance**

Reviewer #1: I have no major objections.

Reviewer #2: Specific comments:

1) The data in Figure 3B shows significant differences in the kinetics of the responses, but not whether the effector function at any timepoint is different. The conclusion that M09-specific cells are functionally superior would be strengthened by comparing the frequency tetramer+ cells at each timepoint, with those that produce cytokines. Or, alternatively/in addition, by the ratio of IFNg only producers, to producers of 2 or 3 cytokines.

2) The data in Figure 4 show differences in effector function at a transcriptional level, but at a single timepoint. This may be misleading since the M09 cells are peaking at d40, while M25-specific cells have long-since contracted. I wondered how the comparison of T cell function between M09-specific vs other T cells in Figure 3 would differ if compared at the peak of expansion.

3) Even if the M09-specific T cells are more functional on a per cell basis, I’m not sure how the vaccination data should be interpreted. There is currently no data to suggest that the vaccination induces M09-specific cells with altered effector function. Unless there is something special about the M09 peptide that promotes functionally distinct T cells, it would seem most likely that any improved function of M09-specific T cells is due to specifics of the MCMV infection context, which would be normalized by the vaccination platform. Nevertheless, vaccination with the M09 epitope reduced acute MCMV replication compared to other epitopes (Fig 5). Therefore, it seems possible (or maybe likely) that the efficacy of M09-specific T cells is related to the target, either as a superior epitope, or due to some unique feature of MCMV infected targets and how they express and present the M09 protein or epitope compared to other epitopes.

4) The frequencies and numbers of M25 and M142-specific T cells assessed by tetramer in figure 1 don’t seem to parallel the data obtained by cytokine production in Figure 3. The cytokine data shows a much more prominent expansion and peak of the response at d8, followed by a sharp contraction compared to tetramer staining shown in 1C. It is hard to directly compare the data in these two figures. It would be helpful to show how tetramer+ staining compared with cytokine+ cells from the same mouse to know whether the tetramers or the cytokines were under or over-reporting on the cell numbers.

Reviewer #3: (No Response)

**Part III – Minor Issues: Editorial and Data Presentation Modifications**

Reviewer #1: 1. Figure 2D – although the data are nice and go hand in hand with other results, one wonders whether mutations in K181 M09 would diminished tetramer binding and thus mask real M09-specific CD4 T cell response in mice infected with MCMV K181 and Smith M09K? If possible, authors should make a tetramer based on K181 M09 sequence and repeat staining. The result would not diminished their conclusions and the functionality of M09-specific CD4 T cells, however, it would provide relevant information.

2. Do M09Smith and M09K181-specific CD4 T cells differ in their functionality at early (day 8) and late (day >40) time points? The functionality of CD4 T cells specific for both Smith and K181 M09 should be explored and tested for IFNg, TNFa and IL-2 production.

3. Line 151 and Fig S2A and B – graph shows viral titer in spleen and liver from mice infected with MCMV Smith and Smith M09K only. Authors should also include viral titers of MCMV K181 and K181 M09S (as they mentioned in the text).

4. Figure S2C and D – the kinetics of MCMV K181 and K181 M09S might be shown also.

5. Figure 2D – include mock values

6. Although the data are nicely presented, the reader would benefit if more information on the graphs are shown; such as a type of tested organ (for example 1B, 1C, 1D, Figure 3, etc) or a legend (4A).

7. Line 146 – authors should clarify this better (I guess authors mean epitopes M25, M142 and M09)

8. Figure 1B – one might find the layout of the graphs confusing

9. Figure 1D – population percentage font size in dot plots should be bigger

10. Line 138 – the dot should be removed

Reviewer #2: 1) It appears that the data showing the Smith and K181 titers in the SG in Figure 2A and 2C are identical. If so, and the authors feel that it is essential to present the data this way, they should indicate this duplication in the legend. However, it appears to me that the data as presented in Fig 2C is sufficient to draw the conclusions as written in the text and 2A could be deleted unless it is from a distinct experiment.

2) Please add information about how viral stocks were generated and maintained (i.e. tissue culture- or SG-derived virus?).

3) Fig 3A – The authors discuss in the text (lines 170-171) the frequencies of IL-2 producing T cells as a proportion of M09 or M25-specific T cells, but I believe they are referring to the frequencies of IL-2 or IFNg producers among all CD4s, which is a different conclusion.

Reviewer #3: Fig 1D is not referenced in the results text and the experiment/results are not detailed in the results.

Fig 2A shows MCMV persistence out to day 55 in the SG, have mice infected with K181 been studied longer than this, does the virus remain persistent or is it eventually controlled at much later times?

Fig 3

Could the authors comment on the apparent frequency disconnect between the tetramer data in Fig 1 and the frequency of M09 peptide cytokine producing cells. Fig 1C shows that approx. 10% of the CD4 cells in the SG are M09 positive while specific cytokine production was seen in only 0.2-0.5% Fig 3 B, C,D? Was a marker of activation also used in these experiments and if so are a large proportion of peptide activated CD4 T cells that are not making any of the cytokines measured? Have the M09 specific T cells been assessed for cytotoxic activity?

Fig 4

The results show in Fig 4 and described in the text would benefit from some revision to help with understanding and clarity.

Fig 4A I think the CD44/69/43 and 27 expression levels have been determined on M25 and M09 tetramer positive CD4 T cells but this is implied from the SFig3 figure legend, but is not stated in the Fig 4A figure legend, this could also be stated in the results eg line 180 M25 “tetramer specific” cells and line 186 M25 and M09 “tetramer specific” cells. The flow data in this figure using blur/red grey dashed and solid lines is very difficult to see.

Fig 4D/E please confirm this was tetramer staining of GFP positive (this IL-10 secreating) CD4 cells in SG.

Fig 4E flow plot of IL-10 v CD43 the data is difficult to see as the percentage positive cells text obscures the upper right quadrant of the plot.

Fig 4F Y axis, should I think, be logs 103 to 105 PFU

Fig 5

Fig 5A The “conv” vaccine preparation is a mixture of M25+M139+M142, M09 is added where the vaccines are compared. Results for M138 single vaccination are not show? In addition, are similar levels of the individual T cell responses induced when the combination of conv or conv+M09 are used as the vaccine (measuring individual responses by tetramer but administering the combined antigens) as this is the experimental vaccine condition being investigated?

Fig 5D While there is statistical significance when +/- anti IL-10R conditions are compared, for the key comparison at the 8 and 15 day time points, statistics are not shown for this, so it is unclear if it does test as significant please clarify? However, the point is well made that 3 of 4 and 2 of 4 animals in the CONV+M09 group completely controlled virus.

The title omits that this is murine cytomegalovirus and that the persistence is in the SG?

While the discussion does mention the importance of CD4 cells in human and rhesus cytomegalovirus and suggests that vaccine strategies should consider CD4 T cell generation the implication from this MCMV work is that the peptide in M09 is somewhat unique in generating these late-rising T cells that control persistent replication and that this phenomenon might be mouse strain specific as well as wildtype MCMV isolates having mutations that do not lead to the generation of these T cells. Could the authors expand on there comments to give an idea of how unique the phenomenon might be to the B6 mouse strain? It is also somewhat difficult to imagine how the studies in HCMV CD4 T cells might be performed and maybe the authors could comment on this as well in their discussion.

PLOS authors have the option to publish the peer review history of their article (what does this mean?). If published, this will include your full peer review and any attached files.

Reviewer #1: No

Reviewer #2: No

Reviewer #3: No
---

## [Editor Report · Decision Letter 1]

21 Nov 2023

Dear Chris,

We are pleased to inform you that your manuscript 'Late-rising CD4 T cells resolve mouse cytomegalovirus persistent replication in the salivary gland' has been provisionally accepted for publication in PLOS Pathogens.

Best regards,

Ian R Humphreys

Guest Editor

PLOS Pathogens

Patrick Hearing

Section Editor

PLOS Pathogens

Kasturi Haldar

Editor-in-Chief

PLOS Pathogens

orcid.org/0000-0001-5065-158X

Michael Malim

Editor-in-Chief

PLOS Pathogens

orcid.org/0000-0002-7699-2064

The authors have provided additional data that has strengthened the manuscript, particularly in relation to the functionality of m09-specific CD4+ T cells (including timecourse data). Other responses to the reviewers' comments were reasonable and some issues will be addressed by follow-up larger studies that are beyond the scope of the current manuscript. The conclusions made by the authors are supported by the data provided, and this is a very interesting and well-conducted study.
---

## [Editor Report · Acceptance letter]

1 Dec 2023

Dear Dr. Benedict,

We are delighted to inform you that your manuscript, "Late-rising CD4 T cells resolve mouse cytomegalovirus persistent replication in the salivary gland," has been formally accepted for publication in PLOS Pathogens.

Best regards,

Michael Malim

Editor-in-Chief

PLOS Pathogens

orcid.org/0000-0002-7699-2064